# Pathogen effectors and plant immunity determine specialization of the blast fungus to rice subspecies

Jingjing Liao[1,2†], Huichuan Huang[1,2†], Isabelle Meusnier[3], Henri Adreit[4], Aurélie Ducasse[3], François Bonnot[4], Lei Pan[1,2], Xiahong He[1,2], Thomas Kroj[3], Elisabeth Fournier[3], Didier Tharreau[4], Pierre Gladieux[3], Jean-Benoit Morel[3*]

[1]State Key Laboratory for Conservation and Utilization of Bio-Resources in Yunnan, Yunnan Agricultural University, Kunming, China; [2]Key Laboratory of Agro-Biodiversity and Pest Management of Education Ministry of China, Yunnan Agricultural University, Kunming, China; [3]Institut National de la Recherche Agronomique, UMR BGPI, Montpellier, France; [4]Centre de coopération internationale en recherche agronomique pour le développement, UMR BGPI, Montpellier, France

**Abstract** Understanding how fungi specialize on their plant host is crucial for developing sustainable disease control. A traditional, centuries-old rice agro-system of the Yuanyang terraces was used as a model to show that virulence effectors of the rice blast fungus *Magnaporthe oryzaeh* play a key role in its specialization on locally grown indica or japonica local rice subspecies. Our results have indicated that major differences in several components of basal immunity and effector-triggered immunity of the japonica and indica rice varieties are associated with specialization of *M. oryzae*. These differences thus play a key role in determining *M. oryzae* host specificity and may limit the spread of the pathogen within the Yuanyang agro-system. Specifically, the AVR-Pia effector has been identified as a possible determinant of the specialization of *M. oryzae* to local japonica rice.

**\*For correspondence:** jean-benoit.morel@inra.fr

[†]These authors contributed equally to this work

**Competing interests:** The authors declare that no competing interests exist.

## Introduction

Understanding the mechanisms determining host range of plant pathogens is crucial for disease management strategies, phytosanitary regulations and policies. The recurrent emergence of new pathogen lineages specialized to novel plant species or newly bred resistant varieties is a major limitation to agricultural production, and there is tremendous interest in developing sustainable strategies to prevent pathogen emergence and spread (*McDonald, 2010*; *Giraud et al., 2010*). Finding durable methods of controlling the host range of pathogens requires the understanding of the molecular and physiological determinants of pathogen variation in fitness across space and hosts (*Williams, 2010*; *Barrett et al., 2008*). Before the advent of molecular genetic methods, classical studies in plant pathology have documented patterns of pathogen fitness on different hosts, including pathogenicity (the capacity to infect) and virulence (the quantity of symptoms) (*Johnson, 1961*; *Nadler, 1995*; *Brown, 1994*). Variations in pathogen fitness have been repeatedly investigated for numerous agricultural pathosystems using controlled cross-inoculation experiments or inoculation on series of differential hosts. Numerous studies have reported evidence for pathogen local adaptation, where local pathogens have a greater average fitness on their local hosts than immigrants (*Kaltz and Shykoff, 1998*; *Laine and Barrès, 2013*). Higher pathogen fitness on hosts living in the same habitat is consistent with evolutionary theory, which predicts that parasites should be ahead of

**eLife digest** Microbes that cause diseases in plants are a threat to food security. For example, the rice blast fungus *Magnaporthe oryzae* causes the loss of enough rice to feed 60 million people each year. Disease-causing microbes must overcome the plant's first line of defense, which includes preformed barriers and antimicrobial responses that are triggered by characteristic molecules found in many different microbes.

The microbes that can overcome this first line of defense typically do so with an arsenal of proteins called effectors that interfere with specific biological processes in the plant. To counteract this interference, some plants have evolved genes that encode proteins that detect these effectors and trigger stronger antimicrobial responses. For centuries, farmers and plant breeders have selected for these resistance genes when trying to breed crops that are more resistant to disease. However, over time, disease-causing microbes have lost effectors, which means that several resistance genes have rapidly become ineffective.

Some researchers predicted that growing a mixture of varieties of a given crop together might be a better way of protecting crop yields. Over 16 years ago, this idea was proved successful against the rice blast fungus for rice plants grown in China. However, the exact reasons why this strategy worked and its effects on the fungus were not clear.

Now Liao, Huang et al. have taken another look at rice varieties grown via the traditional method of terraces of rice paddies in Yuanyang. Some of these varieties had a strong first line of defense and few resistance genes, while others relied much more on resistance genes to protect themselves again the rice blast fungus. Liao, Huang et al. found that growing rice varieties with such different immune systems forces some of the rice blast fungi to accumulate effector proteins to combat the first line of defense, whereas other fungi had to get rid of these effectors to avoid being recognized by the major resistance genes. These two forces led to the evolution of two specialized populations of fungi that can infect specific rice varieties but not others. This means that the fungi cannot spread in the landscape, and so the fields of rice become resistant as a whole.

These new findings demonstrate the importance of diversity in rice for sustainable crop protection. The next challenge will be to demonstrate if a similar approach can also protect other major crops grown in different agricultural settings.

their hosts in the co-evolutionary race due to their higher mutation rates, shorter generation times and huge populations sizes (*Gandon and Michalakis, 2002*). Trade-offs among pathogen fitness traits (e.g. between pathogenicity and transmission success rate) are also frequently invoked in theoretical models to explain the maintenance of variation in pathogenicity and resistance (*Brown and Tellier, 2011*; *Laine and Tellier, 2008*). However, although it is important to elucidate the origin and maintenance of variations in pathogen fitness on different hosts for developing durable means of controlling disease, most current understanding is still largely based on theoretical predictions (*Brown and Tellier, 2011*; *Laine and Tellier, 2008*; *Schulze-Lefert and Panstruga, 2011*; *Thrall et al., 2015*). Thus there is a lack of studies investigating the molecular or physiological bases of variation in pathogen fitness across pathogen populations, especially in fungi (*Poppe et al., 2015*).

Our current knowledge about the genetic basis of fungal pathogen specialization determining host range is mainly based on comparative genomics and functional analyses of candidate genes. These studies revealed the pivotal role of effector proteins that are secreted during infection and target cellular processes of the host to promote infection. In plant pathogenic fungi, the most prominent class of effectors are small secreted proteins. They are believed to be mostly involved in the suppression of host immunity and in particular so-called pattern-triggered immunity activated by conserved microbial molecular patterns, such as fungal cell wall components (*Lo Presti et al., 2015*). Comparative genomics have revealed distinct repertoires of effectors between related pathogens specialized on different hosts. This suggests that variation in the composition of pathogen effector repertoires contributes to variation in pathogen fitness on different hosts (reviewed in [*Schulze-Lefert and Panstruga, 2011*]). In the case of *Magnaporthe oryzae* for instance, pathogenicity toward

rice was correlated with the presence of certain effectors (*Chiapello et al., 2015*). The role of variation in pathogen effector repertoires in pathogen specialization is supported by the fact that dispensable, lineage-specific chromosomes containing effectors appear to control adaptation to hosts in a number of fungal plant pathogens (e.g. [*Ma et al., 2010*]). In the rice blast fungus *M. oryzae*, the role of effectors in specialization is supported by circumstantial evidence stemming from the comparisons of isolates specialized to rice and Setaria millet. While the effector-coding gene *AVR1-CO39* was absent from rice-infecting isolates, transgenic expression of *AVR1-CO39* rendered rice isolates non-pathogenic onto rice carrying the appropriate immune receptor (*Couch et al., 2005*). Similar situations have been reported for the *PWL2* gene that prevents pathogenicity on weeping lovegrass (*Sweigard et al., 1995*) and the *Pwt3* and *Pwt4* genes that prevent pathogenicity on wheat (*Takabayashi et al., 2002*). Thus, whereas the role of effectors in pathogenicity has been demonstrated in several cases (for review [*Asai and Shirasu, 2015*]), their role in variations in host range among pathogen populations remains largely unknown. In some cases, certain effectors can be recognized by plant immune receptors (often called resistance proteins), leading to the activation of the so-called effector-triggered immunity. The effectors revealed by their activity rendering some isolates non-pathogenic ('avirulent') on some hosts represent a sub-category later called *Avr*-effectors (*Jones and Dangl, 2006*). The strong and specific resistance level conferred by effector-triggered immunity contrasts with basal immunity which is weak and not specific and relies on a combination of different mechanisms like constitutive expression of defense genes and pattern-triggered immunity (*Lo Presti et al., 2015*; *Vergne et al., 2010*). The respective roles of effector-triggered and basal immune responses in pathogen host range variations have yet to be investigated.

Rice blast caused by *M. oryzae* is currently the most damaging rice disease worldwide, occurring on all cultivated subspecies and varietal types of rice. Four major lineages of *M. oryzae* causing rice blast can be distinguished on a worldwide scale (*Saleh et al., 2014*). The rice - *M. oryzae* pathosystem is particularly well-suited for studying specialization to the host since a large number of effectors and resistance (*Pi*) genes coding for immune receptors have been cloned and basal immunity is now well-understood (*Azizi et al., 2016*; *Liu et al., 2014*). Moreover, large-scale cross-inoculation experiments of a collection of rice varieties with a collection of rice blast samples representing the worldwide diversity revealed patterns of pathogen fitness that suggest the existence of specialization to hosts in this pathosystem (*Gallet et al., 2016*). Strains originating from japonica rice infected most japonica varieties but could not infect indica varieties whereas strains derived from indica rice infected indica and japonica varieties in controlled conditions. Inoculation onto varieties containing different *Pi* resistance genes suggested an important role of these genes in the observed patterns of pathogenicity. This is due to the fact that strains originating from japonica hosts were able to overcome less resistance genes than strains originating from indica hosts. However, conclusions regarding the determinants of host range and pathogen specialization were hindered by the fact that the plants and fungal isolates tested had not been collected at the same sites, and were not actively involved in co-evolutionary interactions.

In this study, we investigated the molecular basis of *M. oryzae* specialization to its hosts: rice subspecies japonica and indica. To investigate the mechanisms of specialization in *M. oryzae* populations actively co-evolving with their hosts, a traditional agro-system from the Yuanyang terraces (Yunnan, China; http://whc.unesco.org/en/list/1111/) where indica and japonica rice varieties have been grown side-by-side for several centuries (*He, 2011*) was used. First, we showed that pathogen populations are specialized to indica and japonica rice varieties. Next, we investigated the role of plant immunity in shaping variations in pathogen fitness and the contribution of effectors to this pattern. We discovered that specialization of *M. oryzae* isolates to japonica and indica varieties grown in Yuanyang is correlated with, respectively, the deployment of a large number of *Avr*-effectors in japonica-borne isolates (i.e. effectors triggering complete resistance in some plant genotypes) and a large depletion of Avr-effectors in indica-borne isolates. These contrasting effector repertoires mirror the significant immunity differences between japonica and indica local varieties. We provide further evidence that the *AVR-Pia* effector is possibly a key player in the pattern of specialization to indica or japonica rice varieties observed.

## Results

### Differentiation of *Magnaporthe oryzae* populations infecting indica or japonica rice in Yuanyang

A total of 214 *M. oryzae* isolates were collected from rice plants (*Oryza sativa*) between 2009 and 2013 in the Yuanyang terraces where the majority of cultivated rice belongs to the indica sub-species (98% of the ~1000 ha; data from local station). Isolates were sampled on both indica and japonica sub-species (n = 177 and n = 37, respectively) and genotyped using 13 microsatellites. Over this period, the two indica Acuce and Xiao Gu and the two japonica Huang Pi Nuo and Nuo Gu represented the most commonly grown varieties in Yuanyang. Neighbor-joining analysis of genetic distances (*Figure 1A*) combined with DAPC (*Figure 1—figure supplement 1*) circumscribed a single group representing the vast majority (92%) of isolates collected on japonica rice (half on Huang Pi Nuo and one quarter on Nuo Gu varieties) and therefore referred to as the 'japonica-borne' (JB) group. Conversely, 94% of isolates collected on indica rice were not assigned to the JB group, forming four main clusters collectively referred to as the 'indica-borne' (IB) group (*Figure 1A*; *Figure 1—figure supplement 1*). Population genetic analysis based on linkage disequilibrium did not support the existence of regular sexual reproduction (*Figure 1—figure supplement 2*). No clear pattern of association was found between the making up of pathogen clusters and other possible structuring factors such as the year of sampling or altitude (*Figure 1—figure supplement 3*). Thus the rice sub-species appear to be the most potent factor structuring pathogen populations in this agro-system.

We used cross-inoculation assays in controlled conditions to evaluate how far pathogen specialization to host subspecies contributed to the observed pattern of pathogen population subdivision between JB and IB groups. Thirty representative isolates were selected (marked with a '\*' on *Figure 1*) and inoculated onto the varieties most frequently cultivated in the Yuanyang terraces to evaluate qualitatively cross-pathogenicity using a classification of either resistant or susceptible plant (*Figure 1—figure supplement 4*). Japonica rice were susceptible to all isolates from the JB group but only few instances of susceptibility of indica varieties were found with these isolates (*Figure 1C*). By contrast, both indica and japonica varieties were susceptible to most IB isolates (see below). Altogether, these results suggest that JB and IB groups of *M. oryzae* are differentially adapted to japonica and indica varieties grown in Yuanyang.

### Major resistance genes in indica varieties prevent infection by japonica-borne isolates

We investigated the factors preventing infection of indica varieties by JB isolates in both field (*Figure 1B*) and controlled conditions (*Figure 1C*). We first evaluated the role of effector-triggered immunity by assessing the content in terms of major resistance (*Pi*) of Yuanyang rice varieties. We used whole-genome sequencing of four Yuanyang varieties (*Figure 2—figure supplement 1*) and pathogenicity assays (*Figure 2—figure supplement 2*) to allow detection of 19 cloned *Pi* genes. These analyses showed that the two major indica rice varieties Xiao Gu and Acuce harbored 8 and 7 *Pi* genes compared to 2 and 3 in the Huang Pi Nuo and Nuo Gu japonica varieties (*Figure 2A*). In addition, the indica varieties from Yuanyang had a higher content of *Pi* genes compared to indica varieties grown in a non-traditional agro-system nearby (*Figure 2—figure supplement 3*). Interestingly, one of the most frequent *Pi* genes in the indica varieties from Yuanyang was the *Pia* gene that had a low frequency in varieties from the non-traditional agro-system.

Secondly, we tested whether JB isolates were excluded from indica hosts due to multiple *Avr*-effectors matching *Pi* genes and thus conferring 'avirulence'. Thirty representative isolates were selected based on the results of population structure analyses (*Figure 1C*) and inoculated on a set of rice lines diagnostic for 14 major *Pi* genes ([*Berruyer et al., 2003*] and references therein), thus allowing the identification of the 14 corresponding *Avr*-effectors (*Figure 2—figure supplement 4*). On average, isolates from the JB group had 11.9 *Avr*-effectors (*Figure 2B*), which is significantly different from (p<10$^{-7}$) and almost twice as much as in the IB group with 6.5 *Avr*-effectors. These results suggest that the high content of *Avr*-effectors in JB isolates accounts for their lack of pathogenicity on indica varieties that contain many *Pi* genes. Thus multiple gene-for-gene interactions, such as the *Pia/AVR-Pia* interaction, could act individually or in combination to prevent disease.

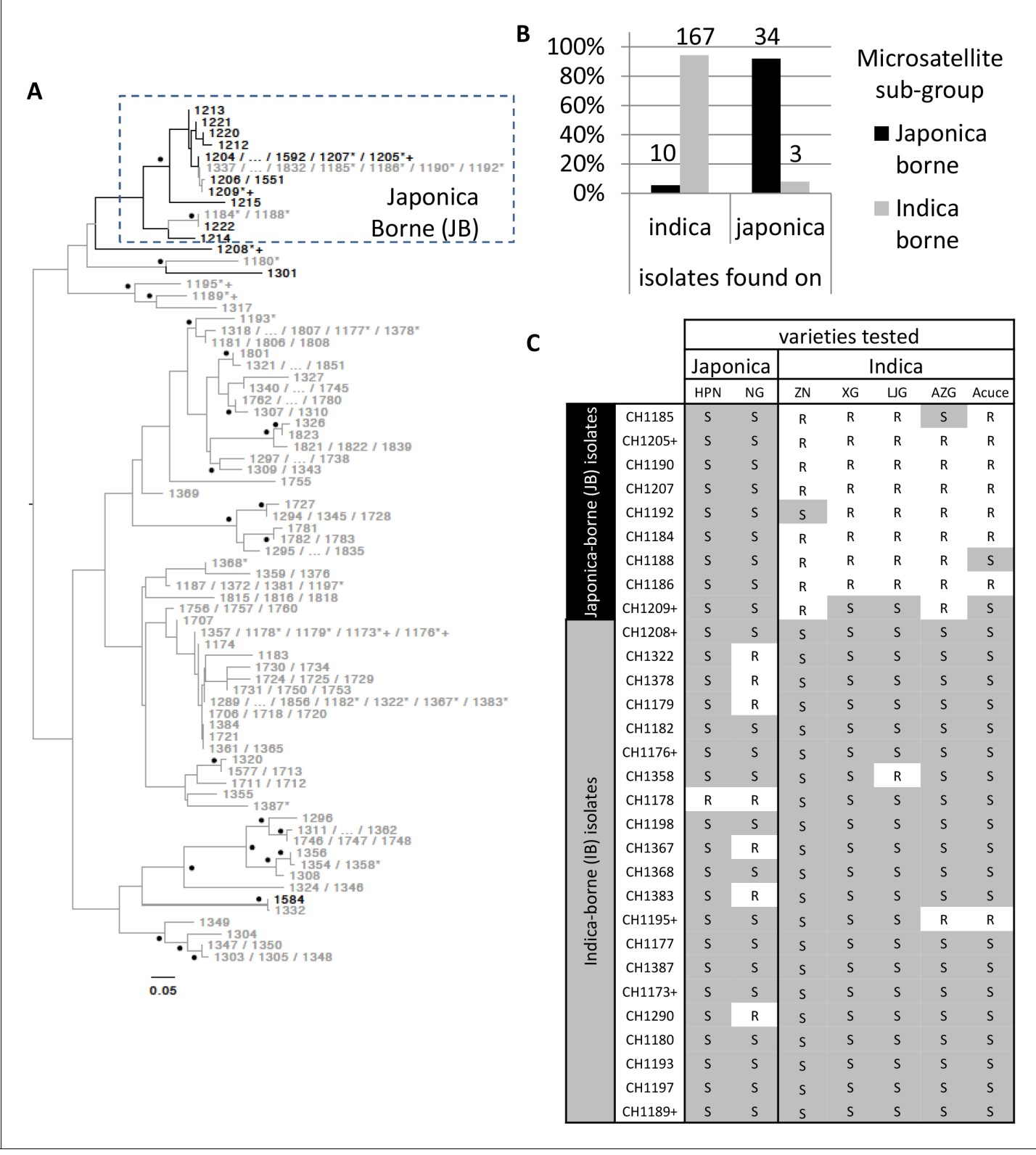

**Figure 1.** Variability in microsatellite genotype and pathogenicity phenotype of *M. oryzae* isolates harvested on indica and japonica rice grown in Yuanyang terraces. (**A**) Midpoint rooted neighbor-joining dendrogram representing the proportion of shared microsatellite alleles among multilocus genotypes. Two hundred fourteen isolates (the prefix 'CH' visible in C was removed from A for clarity) were genotyped using 13 microsatellites. Only one representative of multilocus genotypes repeated multiple times was kept, and for each repeated multilocus genotype the corresponding isolates are listed at the tip of a branch (74 unique multilocus genotypes in total). Bootstrap supports are indicated by a black dot when >40% (1000

*Figure 1 continued on next page*

*Figure 1 continued*

resamplings). Isolates harvested on japonica and indica rice are indicated in black and grey respectively. Six isolates (i.e. CH1180, CH1189, CH1195, and CH1317, collected on indica; CH1208 and CH1301, collected on japonica) that show in the dendrogram an intermediate position, were assigned to the IB group because of their pathogenicity phenotypes and their separation from the other JB genotypes in the DAPC (*Figure 1—figure supplement 1*). The isolates selected for (**C**) and *Figure 3* are marked by a '*' and a '+' respectively. The cluster of Japonica-borne isolates (abbreviated JB; within the square with dashed lines) was defined following Discriminant Analysis of Principal Components; the remaining samples represent the cluster of Indica-borne (IB) isolates. (**B**) Distribution of the two clusters identified based on microsatellite variation (japonica borne 'JB' and indica borne 'IB') on japonica and indica hosts in the Yuanyang terraces. The fact that the distributions are largely non-overlapping (the JB and IB clusters are mostly found on Japonica and Indica hosts, respectively) suggests local adaptation of the pathogens to their respective hosts. The numbers of isolates are indicated above the bars. (**C**) Pathogenicity profiles of 30 isolates on two japonica and five indica varieties ('R' and 'S' stand for resistance and susceptibility, respectively). The 30 representative isolates were selected from the analysis presented in (**A**) and inoculated onto seven major rice varieties grown in Yuanyang (HPN: Huang Pi Nuo; NG: Nuo Guo; ZN: Zi Nuo; XG: Xiao Gu; LJG: Li Jiao Gu; AZG: Ai Zhe Gu; Acuce). HPN, NG and ZN are all glutinous rice varieties. The isolates marked with a '+' in (**A**) and (**C**) are those used in *Figure 3*; all isolates included in (**C**) are marked with a '*' in (**A**). The qualitative analysis of symptoms presented here suggests that japonica-borne (JB) isolates cannot attack indica rice whereas indica-borne (IB) isolates can attack japonica.

The following source data and figure supplements are available for figure 1:

**Source data 1.** The data relates to *Figure 1*.

**Figure supplement 1.** Neighbor-joining tree representing the genetic distance (in terms of proportion of shared alleles) between the 74 unique microsatellite genotypes characterized in Yuanyang (left) and patterns of memberships in K = 2 to K = 10 clusters as inferred using DAPC (right).

**Figure supplement 2.** Summary statistics of genetic variability in the four clusters of *Magnaporthe oryzae* identified using Discriminant Analysis of Principal Components and neighbor-joining analysis of genetic distance (see *Figure 1—figure supplement 1*).

**Figure supplement 3.** Midpoint rooted neighbor-joining dendrogram representing the proportion of shared microsatellite alleles among the 214 multilocus genotypes originating from Yuanyang terraces.

**Figure supplement 4.** Scale used for scoring global incompatibility/compatibility (Resistance/Susceptibility).

## Elevated basal immunity in Yuanyang japonica varieties

The type of lesions caused by IB and JB isolates on japonica rice from Yuanyang were strikingly different (*Figure 3—figure supplement 1*). Lesions caused by JB isolates had a drastically reduced brown halo, a phenotype that is associated with resistance and correlated with the production of reactive oxygen species (*Hayashi et al., 2016*). This indicated that JB isolates may have the capacity to inhibit this important component of basal immunity (*Jones and Dangl, 2006*). This also suggested that IB isolates may face high basal immunity in Yuanyang japonica varieties. To test this hypothesis, we first evaluated basal immunity conferring partial protection in the absence of major resistance genes and relying on a combination of several different molecular processes such as preformed defense (*Vergne et al., 2010*; *Delteil et al., 2012*) and pattern-triggered immunity (*Lo Presti et al., 2015*). We inoculated four broadly infecting isolates that harbor few *Avr*-effectors (*Gallet et al., 2016*) to evaluate resistance under conditions that minimize effector-triggered immunity (*Vergne et al., 2010*). This analysis showed that japonica varieties from Yuanyang are more resistant than indica varieties (*Figure 3—figure supplement 2*), comparable to the level observed in Nipponbare, which is renowned for its elevated basal immunity (*Vergne et al., 2010*).

To analyze preformed defense in Yuanyang japonica varieties, the constitutive expression of defense-related genes which is a hallmark of basal immunity towards the rice blast fungus (*Vergne et al., 2010*; *Delteil et al., 2012*), was determined. When measuring the expression of 16 *PR* genes and 12 genes involved in resistance signaling that are frequently co-regulated with defense genes (see list in *Figure 3—figure supplement 3*), more than half of the genes (16/28) and more than two thirds of the *PR* genes (11/16) showed higher constitutive expression in Yuanyang varieties compared to Nipponbare renowned for its elevated constitutive defense (*Vergne et al., 2010*) (*Figure 3A*), with the Huang Pi Nuo variety showing the strongest constitutive expression of defense-related markers (*Figure 3—figure supplement 4*). In the japonica varieties, in particular Nuo Gu, as opposed to the indica varieties Acuce and Xiao Gu, the constitutive expression of six out

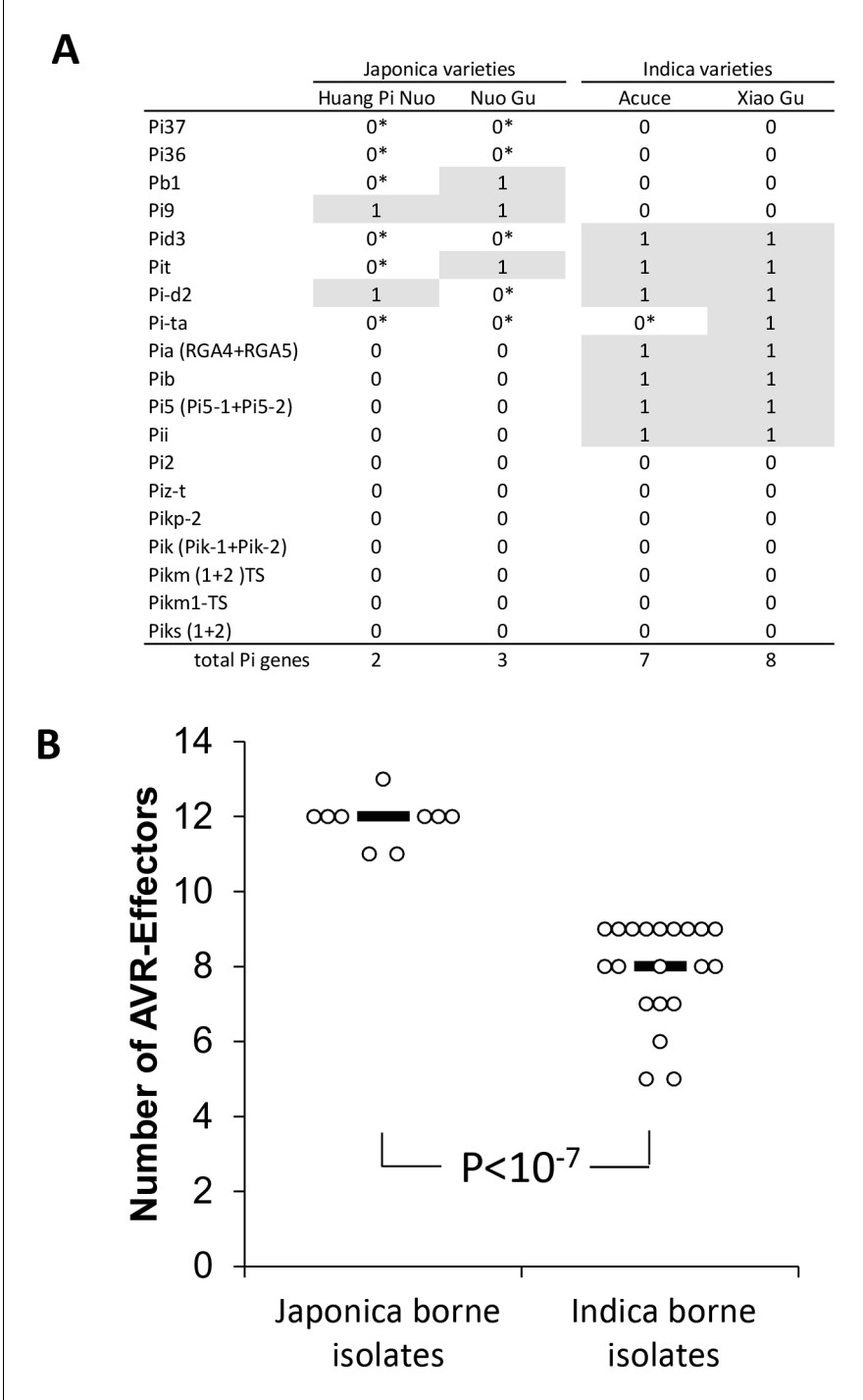

**Figure 2.** Evaluation of rice *Pi* and blast Avr-effectors genes in Yuanyang rice varieties and *M. oryzae* isolates. (**A**) Presence (1)/absence (0) of 19 cloned *Pi* gene based on sequence analysis. Pair-end reads from the four varieties were produced (53 to 58 million reads) by whole-genome sequencing. The reads (~150 nucleotides) were mapped on the corresponding *Pi* sequences using SOAPaligner (http://soap.genomics.org.cn/soapaligner.html) and two mismatches were allowed. Some genes (noted 0*) were present but contained a premature stop codon or were not functional according to inoculation tests (*Figure 2—figure supplement 2*). The sequences used are listed in *Figure 2—figure supplement 1*. Note that when two genes are required for resistance (Pia, Pi5, Pik, Pik-s and Pik-m), the sequences of both genes were analyzed. (**B**) The number of Avr function of 14 effectors was measured (*Figure 2—figure supplement 4*) in the JB and IB groups of 30 isolates defined in *Figure 1C*. Each dot represents

*Figure 2 continued on next page*

*Figure 2 continued*

an isolate and the median (black bar) is indicated. The average values between JB and IB isolates are significantly different ($p < 10^{-7}$; t-test).

The following figure supplements are available for figure 2:

**Figure supplement 1.** Estimation of the presence of Pi genes by mapping reads from whole-genome sequencing on the corresponding Pi gene sequence.

**Figure supplement 2.** Estimation of the presence of the indicated *Pi genes* by inoculation with GUY11 transformed with the cognate Avr-Effector. "1" is presence (plant is resistant), "0" is absence of detection (plant is susceptible).

**Figure supplement 3.** Estimation of the frequency of *Pi* genes in 18 (other than those listed in *Figure 2A*) indica, traditional varieties from Yuanyang terraces (YYT) and in 15 modern, improved indica varieties from the Shiping county (SP; Yunnan, China) used as a local geographical control.

**Figure supplement 4.** Estimation of Avr-Effector complement using rice differential lines in 30 selected isolates from Yuanyang (these isolates are indicated by a '*' on *Figure 1A* showing neutral diversity).

of 16 analyzed *PR* genes was also significantly higher (see Mock treatment in *Figure 3B*; *Figure 3— figure supplement 5A*). Thus, Yuanyang varieties displayed elevated constitutive expression of a large set of defense-related genes, with the japonica varieties Huang Pi Nuo and Nuo Gu showing the strongest levels of expression.

The comparison of *PR* gene expression after infection with the virulent isolate Guy11 in two japonica and two indica typical varieties from Yuanyang also showed a higher induction of defense in the japonica varieties than in the indica varieties (*Figure 3B*; *Figure 3—figure supplement 5B*). This trend was also visible for two other broadly infecting isolates (*Figure 3—figure supplement 6*).

To evaluate the contribution of pattern-triggered immunity to the elevated basal immunity of Yuanyang japonica varieties, we measured the responsiveness to exogenous chitin, a fungal cell wall component inducing immune responses in rice (*Azizi et al., 2016*; *Liu et al., 2014*) and other plants (*Lo Presti et al., 2015*). We compared Huang Pi Nuo and Nuo Gu to Nipponbare which displays high basal immunity (*Vergne et al., 2010*). One day after chitin treatment, the Yuanyang japonica varieties and Nipponbare showed a strong induction of chitin-responsive genes (*Figure 3C*; *Figure 3—figure supplement 7*) with a stronger induction of the *PR* genes in the Yuanyang varieties, particularly in Nuo Gu. Altogether these data indicate that the Yuanyang japonica varieties display hallmarks of an elevated basal immunity compared to Yuanyang indica varieties.

## A large effector complement is required to infect japonica varieties

Since Yuanyang japonica varieties display elevated basal immunity and are infected by JB isolates having a large *Avr*-effector complement, we hypothesized that the large set of *Avr*-effectors in JB isolates was required to counter the basal immunity of japonica varieties. Under this hypothesis, a first expectation was that the number of *Avr*-effectors should be positively correlated with virulence on Yuanyang japonica varieties. Seven isolates from the JB and IB groups with a number of *Avr*-effectors ranging from 5 to 13 (*Figure 1C*) were inoculated on the two major japonica varieties in Yuanyang, Huang Pi Nuo and Nuo Gu. This showed that two components of virulence, aggressiveness (lesion surface) and infectivity (percentage of susceptible lesions per leaf) were strongly correlated with the number of *Avr*-effectors (*Figure 4*).

The second expectation was that the effect of mutations negatively impacting the plant immune system could be reduced when tested with isolates carrying a large effector complement. Indeed a large effector complement may have the capacity to dampen basal immunity down to the low level found in immune-deficient plants. We thus tested the capacity of Yuanyang isolates with different numbers of *Avr*-effectors to infect the *cebip* mutant, defective in mounting chitin pattern-triggered immunity and showing reduced basal immunity (*Delteil et al., 2012*). As previously described, the immuno-depressed plants were less resistant than wild type plants. However, this reduced resistance

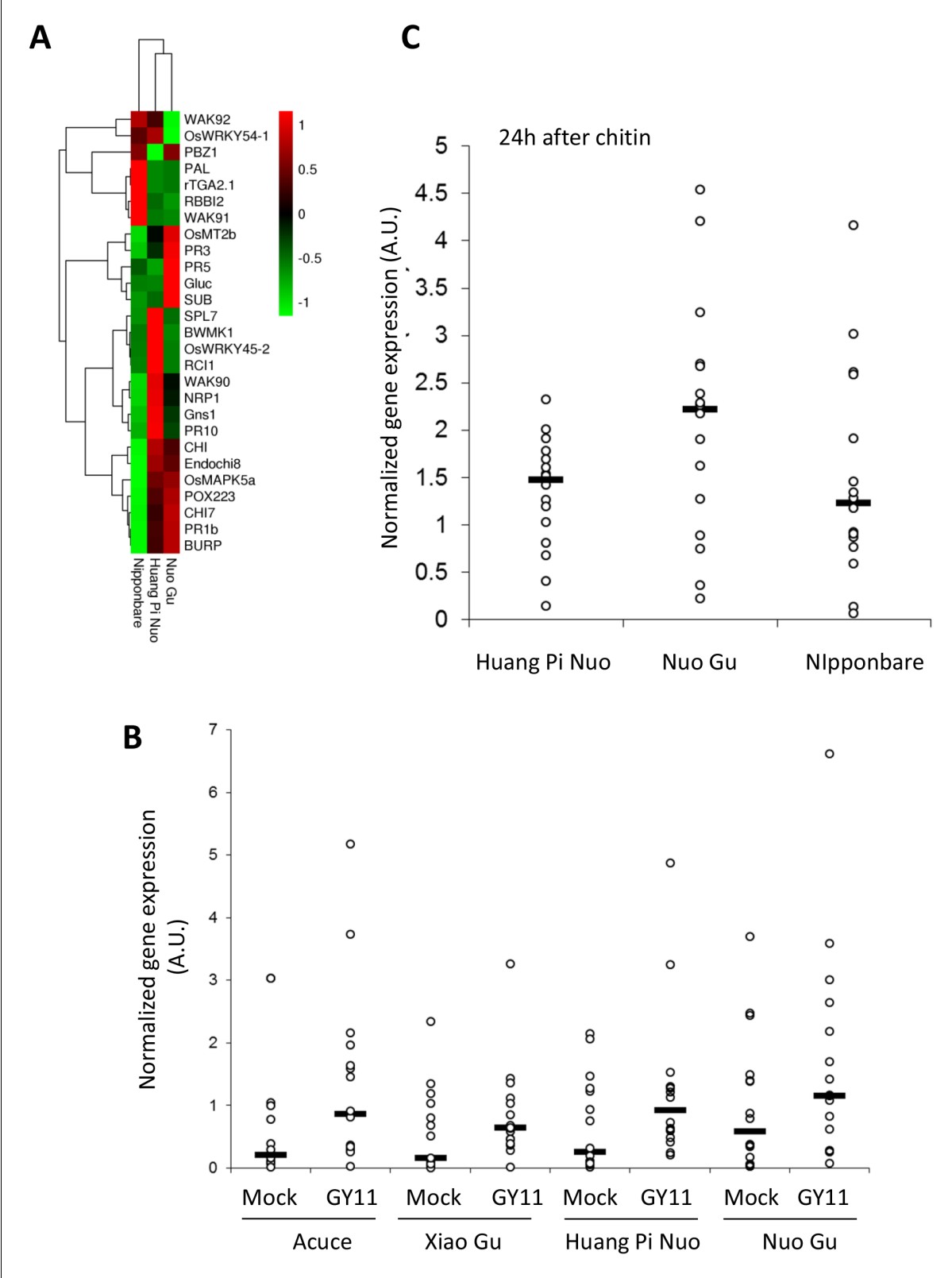

**Figure 3.** Constitutive and inducible defense in Yuanyang rice varieties. The expression of defense-related genes (see *Figure 3—figure supplement 3*) in indica (Acuce, Xiao Gu) and japonica (Huang Pi Nuo and Nuo Gu) varieties grown in Yuanyang as well as the japonica Nipponbare variety was measured by RT-qPCR. In order to make genes with different expression levels comparable, the different values obtained for each gene were normalized by the average value of the considered gene across all measures. (**A**) The constitutive expression of 16 Pathogenesis-related genes and 11
*Figure 3 continued on next page*

*Figure 3 continued*

genes involved in basal immunity signaling was measured by RT-qPCR. The mean values from four biological replicates was normalized and used for hierarchical clustering using hcluster algorithm (www.omicshare.com/tools). The corresponding mean, SD and statistical tests can be found in *Figure 3—figure supplement 4*. (B) The two major indica (Acuce and Xiao Gu) and japonica varieties found in Yuanyang (Huang Pi Nuo and Nuo Gu) were inoculated with the virulent isolate Guy11 or mock treated. The expression of 16 Pathogenesis Related-genes was measured and the mean from four biological replicates was calculated. Each dot represents the average expression value from 16 defense genes. The black bar represents the median value. The corresponding mean, SD and statistical tests can be found in *Figure 3—figure supplement 5*. (C) The expression of 16 Pathogenesis-related genes was measured 24 hr after chitin (100 μg/mL) treatment. The mean value from four biological replicates was calculated and each dot represents this value for one gene. The corresponding mean, SD and statistical tests can be found in *Figure 3—figure supplement 7*.

The following source data and figure supplements are available for figure 3:

**Source data 1.** The data relates to *Figure 3*.

**Source data 2.** The data relates to *Figure 3—figure supplement 2*.

**Source data 3.** The data relates to *Figure 3—figure supplement 4*.

**Figure supplement 1.** Examples of symptoms on Huang Pi Nuo.

**Figure supplement 2.** Average susceptibility of Yuanyang terraces varieties.

**Figure supplement 3.** Accessions and names of genes used for expression analysis.

**Figure supplement 4.** Constitutive expression of defense-related genes in Yuanyang japonica and Nipponbare varieties.

**Figure supplement 5.** Constitutive and fungal-induced expression of pathogenesis-related genes in Yuanyang japonica and indica varieties.

**Figure supplement 6.** Constitutive and fungal-induced expression of pathogenesis-related genes in Yuanyang japonica and indica varieties.

**Figure supplement 7.** Chitin-induced expression of defense-related genes in Yuanyang japonica and Nipponbare varieties.

was only observed with isolates containing a small *Avr*-effector complement but not with JB isolates containing a large *Avr*-effector complement (*Figure 4—figure supplement 1*). These results indicate that basal immunity has a strong negative impact on the virulence of isolates harboring a relatively limited set of *Avr*-effectors but not of isolates with a large *Avr*-effector arsenal. Overall, our findings hence validate the hypothesis that the *Avr*-effector complement in JB isolates is critical for dampening the basal immunity of Yuanyang japonica rice.

## *AVR-Pia* contributes to virulence toward some japonica rice varieties

The best candidates for individual *Avr*-effectors with a substantial impact on virulence on Yuanyang japonica rice were AVR-Pia and AVR-Pii since both are present in almost all JB isolates and completely absent from IB isolates (*Figure 2—figure supplement 4*). Since numerous effectors in JB isolates could have redundant functions, we reasoned that deleting *AVR-Pia* or *AVR-Pii* from these isolates may not strongly affect virulence. We therefore decided to test the role of these effectors by transferring them into the AVR depleted strain Guy11 (26). Only *AVR-Pia* showed significant effects (*Figure 5—figure supplement 1*) that were further confirmed by comparing three independent transgenic strains expressing *AVR-Pia* and three independent strains containing the empty vector on several japonica rice varieties (*Figure 5*). Two key parameters of virulence, the percentage of susceptible lesions and the surface of individual lesions, were significantly higher for the *AVR-Pia* transgenic strain and the effect of *AVR-Pia* was most significant on the Yuanyang variety Huang Pi Nuo. This indicates that *AVR-Pia* makes a significant contribution to the virulence of *M. oryzae* on this particular japonica variety, potentially by interfering with cellular host processes important for infection. However, *AVR-Pia* alone was not sufficient to increase virulence on the Nuo Gu japonica variety, as expected due to the strong inducibility of pattern-triggered immunity in this variety (*Figure 3C*).

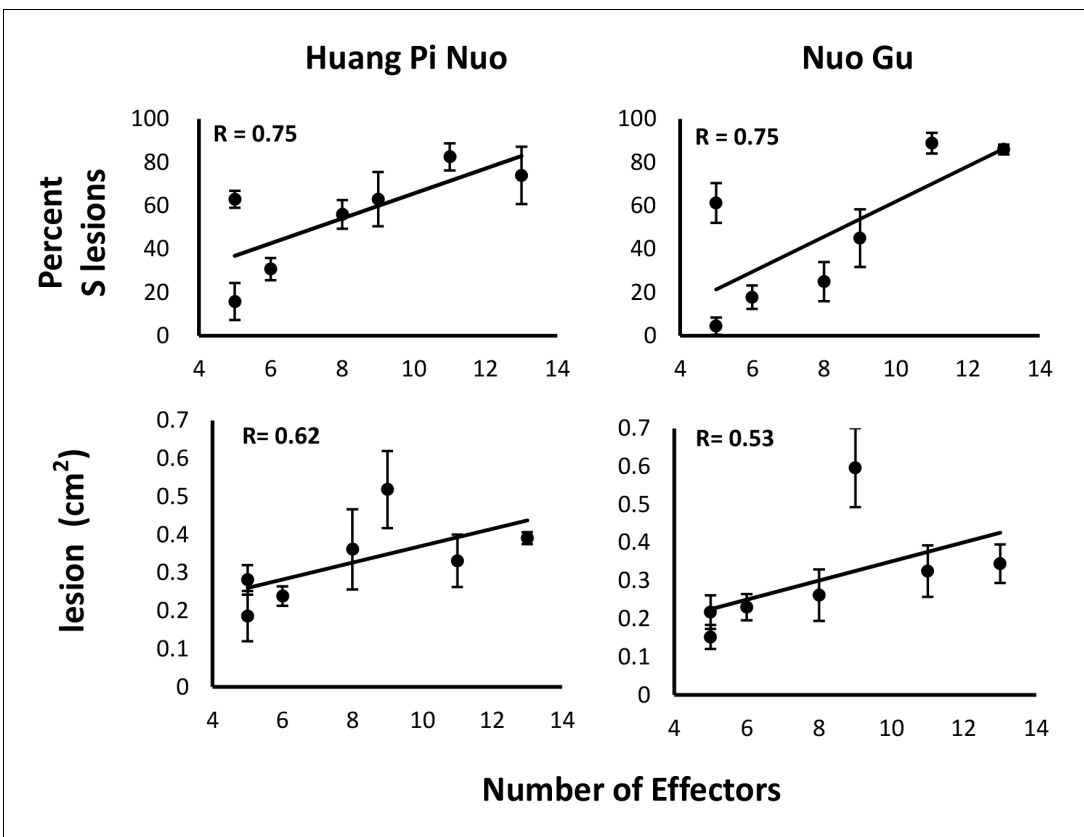

**Figure 4.** Effector complement and virulence of Yuanyang isolates. Seven isolates with Avr-Effector number ranging from 5 to 13 (see isolates marked with '+' in *Figure 1*) were selected and inoculated onto japonica varieties Huang Pi Nuo and Nuo Gu. The two quantitative components of virulence (lesion surface and percentage of susceptible lesions over total lesion number) are indicated as means and standard deviation from three biological replicates (each replicate included six independent plants).

The following source data and figure supplement are available for figure 4:

**Source data 1.** The data relates to *Figure 4*.
**Source data 2.** The data relates to *Figure 4—figure supplement 1*.
**Figure supplement 1.** A large complement of effector is no required on immune-deficient plants.

## Discussion

### Specialization of *M. oryzae* to its hosts in a long-lasting, traditional farming system

The specialization of the rice blast fungus to indica or japonica varieties has been occasionally suggested in the literature but has never been documented convincingly in a real agro-system where both types of hosts occur (*Gallet et al., 2016*; *Shang et al., 2016*). Most notably, previous studies suffered from the use of fungal isolates and plant genotypes that were un-paired, i.e. not obtained from the same plant or area. This therefore did not allow the clear detection of the differential adaptation of pathogens to their local hosts. In our study, we used a large collection of isolates (>200) and their paired rice hosts to investigate the specialization of *M. oryzae* to the host genotype (*Figure 1*). The population of *M. oryzae* in the traditional agro-system of the Yuanyang terraces is highly diverse and structured (*Figure 1* and *Figure 1—figure supplements 1–3*). In particular, there is strong genetic differentiation between isolates from rice host plants belonging to the indica or the japonica subspecies (IB and JB isolates respectively). Interestingly, IB isolates that were rarely

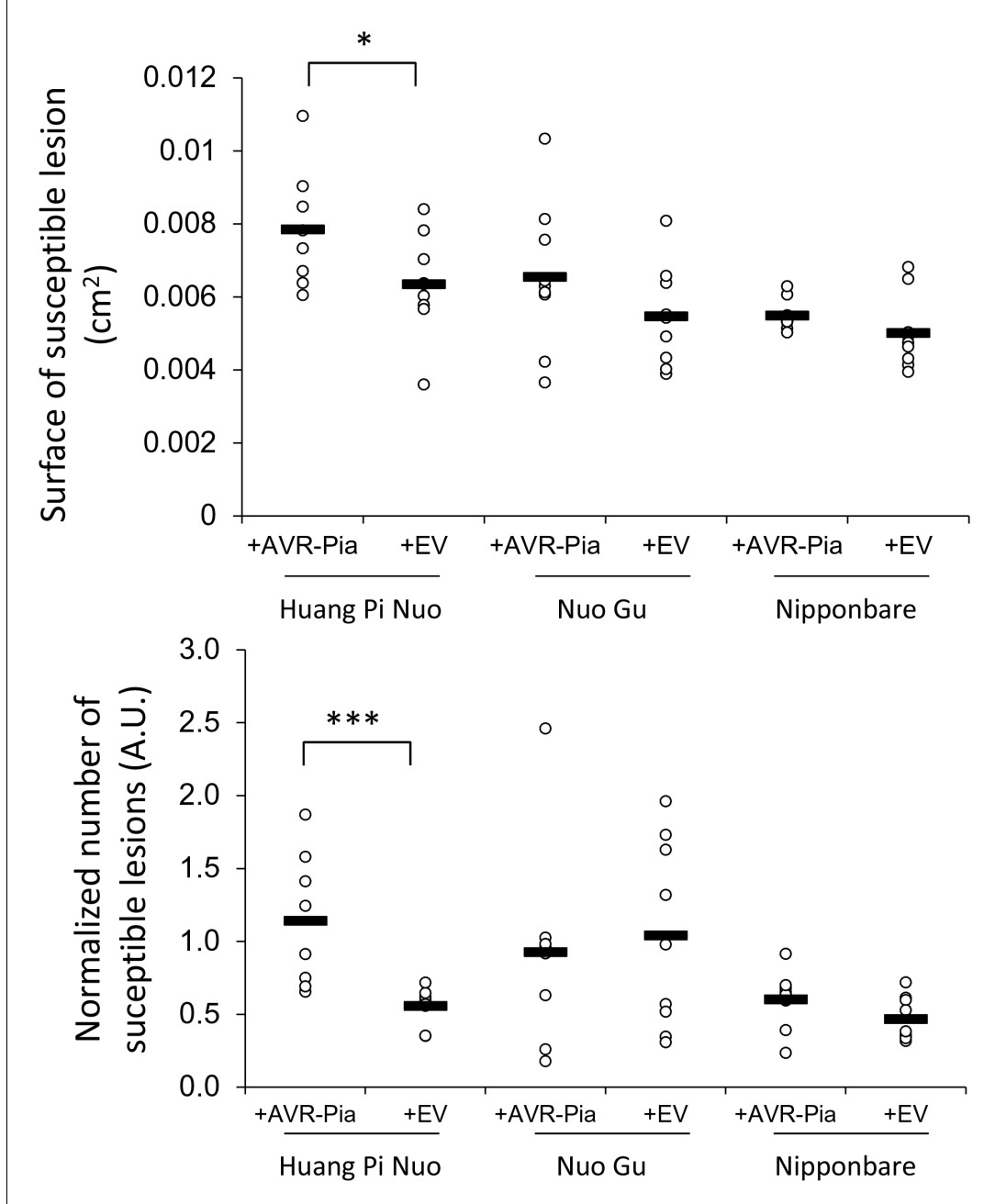

**Figure 5.** Impact of the *Avr-Pia* gene on the virulence of *M. oryzae* on Yuanyang japonica rice varieties. Three independent transgenic isolates expressing the *Avr-Pia* gene under its native promoter in the Guy11 background (+Avr-Pia) or three transgenic containing an empty vector (+EV) were inoculated onto the two major japonica varieties grown in Yuanyang (Huang Pi Nuo and Nuo Gu) and two other japonica variety Nipponbare. On Huang Pi Nuo, *Avr-Pia* isolates are significantly different from isolates containing an empty vector (ANOVA followed by T-test; *p<0.05; ***p<0.001) for two parameters of virulence, the percentage of susceptible lesions (lower panel) and the surface of individual lesions (upper panel). Bars represent average values based on three biological replicates for each of the three *Avr-Pia* and control strains.

The following source data and figure supplement are available for figure 5:

**Source data 1.** The data relates to *Figure 5*.

**Figure supplement 1.** The *Avr-Pia* but not *Avr-Pii* gene affects virulence.

sampled on japonica rice in the field were pathogenic on japonica varieties from the Yuanyang terraces or other origins in controlled conditions (*Figure 1B*). However, they showed reduced virulence on japonica rice that was only detected when quantitative differences were recorded in infection experiments under controlled conditions (*Figure 4*). This reduced virulence of the IB isolates may explain their exclusion from japonica rice in the Yuanyang agro-system. Recently, it has been shown that isolates of *Pyrenophora teres* f. *teres* cause ~9% more lesions on their local barley hosts than immigrants (*Rau et al., 2015*), a value similar to what we observe (*Figure 5*). Contrasting with the pattern uncovered for IB isolates, JB isolates were non-pathogenic on indica varieties under controlled conditions and are largely excluded from such hosts in the field. Therefore, we found clear evidence that, in the Yuanyang terraces, *M. oryzae* isolates are specialized to indica or japonica hosts and that fitness differences on these hosts correlates with genetic differentiation, indicating that there is local, host-driven adaptation in the rice-*M. oryzae* pathosystem. The situation observed in the Yuanyang agro-system is reminiscent of what has been described for several natural ecosystems (for review [*Greischar and Koskella, 2007*]) and represents one of the very few examples known thus far of specialization of a pathogen to its host in an agro-system.

## Plant immunity shapes effector complements and specialization to hosts of the rice blast pathogen

Our data enables the development of a model that can explain the specialization of *M. oryzae* to japonica or indica rice varieties grown in Yuanyang. We demonstrated that the Japonica varieties from the Yuanyang terraces harbor an elevated basal immunity (*Figure 3* and *Figure 3—figure supplements 3–7*) but a low content in major resistance (*Pi*) genes (*Figure 2A*). The larger effector repertoire of Japonica-infecting isolates (*Figure 2B*) could enable them to overcome the elevated basal immunity in Yuanyang japonica varieties (*Figure 3*). In parallel, the relatively larger effector complement of JB isolates would lead to a strong fitness cost on Yuanyang indica varieties (e.g. *Figure 5—figure supplement 1*) that contain a large set of *Pi* genes (*Figure 2A*), with many effectors acting as *Avr*-effectors (i.e. effectors triggering complete resistance in plant genotypes carrying a *Pi* genes that recognize them). By contrast, indica-infecting isolates lack multiple effectors, including *Avr*-effectors, and are therefore able to escape detection by matching *Pi* genes in indica varieties. However, the lack of these effectors induces a strong fitness cost on japonica varieties (*Figure 4*) that have a higher level of basal immunity (*Figure 3*). This model explains why japonica rice varieties, which represent only 2% of the rice grown in Yuanyang, are almost exclusively infected by the particular type of JB isolates and not by the IB isolates that are very much dominating in terms of population size and are able, at least under controlled conditions, to infect japonica rice. In their description of a unifying concept for non-host resistance, host range and pathogen speciation, Schulze-Lefert and Panstruga (*Schulze-Lefert and Panstruga, 2011*) proposed that pattern-triggered immunity and effector-triggered immunity both contribute to non-host resistance, while effector-triggered immunity was proposed to mainly drive host range, i.e. the range of host genotypes that can be infected within a given host species. Our results suggest that several components of basal immunity (including pattern-triggered immunity and constitutive expression of defense) are also important determinants of host range in an agronomical context. Basal immunity in japonica leads to the accumulation of effectors that are detrimental on indica since varieties of this latter rice subspecies display strong effector-triggered immunity thanks to their extended repertoires of immune receptors for *Avr*-effectors.

## Effectors are required for host specialization in the rice/rice blast system

Effectors, fitness cost and differential adaptation are linked in different theoretical models *Giraud et al., 2010*; *Laine and Barrès, 2013*; *Barrett et al., 2009*] and references therein). On the one hand, positive and negative fitness costs of effectors were demonstrated in rare field studies involving the fungus *Leptosphaeria maculans* (*Huang et al., 2006*, *2010*) or the bacterium *Xanthomonas oryzae* pv *oryzae* (*Vera Cruz et al., 2000*). Under controlled conditions, a reduced number of 'avirulence' activities had a significant negative impact on fitness as in the case of *Phytophthora infestans* infecting potato (*Montarry et al., 2010*). This is similar to what we observe with IB isolates that lost many *Avr*-effectors (*Figure 2B*), probably to become infectious on indica varieties. At the

same time, this loss of *Avr*-effectors may negatively impact fitness on japonica rice on which larger repertoires of *Avr*-effectors are required (*Figure 4*). On the other hand, fitness cost and specialization have been observed in many instances (reviewed in [*Laine and Barrès, 2013*]). By contrast, there are yet only few cases demonstrating the relationship between effector suites and specialization in plants or in animals. In plants, a major support for the relationship between effector content and specialization to host plants comes from experiments where the transfer of an entire mobile chromosome harboring effectors could convert a non-pathogenic strain of *Fusarium oxysporum* into a pathogen of tomato (*Ma et al., 2010*). In animals, a convincing example was provided in the case of the transfer of an effector from *Coxiella burnetii* that could extend host cell range of *Legionella pneumophila* (*Lührmann et al., 2010*). Our data suggest that the JB isolates have a characteristic effector suite (*Figure 2B*) that is required for specialization to japonica rice in Yuanyang (*Figure 1*). Our data also suggests that, amongst them, *AVR-Pia* is a major determinant of specialization to either indica or japonica rice subspecies found in the Yuanyang terraces. Indeed we show that (i) the sole presence of *AVR-Pia* increases several fitness parameters (*Figure 5*) on the japonica rice Huang Pi Nuo that has elevated basal immunity (*Figure 3*) and (ii) that *AVR-Pia* is present in almost all JB isolates and absent from IB isolates (*Figure 2—figure supplement 4*). Additional experiments, such as mutating *AVR-Pia* in JB isolates, could provide further insights into the role of this gene in differential adaptation to indica and japonica varieties. To our knowledge this is also the first report in *M. oryzae* of an *Avr*-effector demonstrated to confer increased fitness. Interestingly, despite the gain of fitness provided by *AVR-Pia* on some japonica variety, this gene was lost at a high frequency in *M. oryzae* strains at the worldwide level (*Yoshida et al., 2009*; *Cesari et al., 2013*). This suggests that the *Pia* resistance gene was in the past widely distributed and therefore counter-selected strains harboring *AVR-Pia*. The findings reported here allowed us to propose a model describing the molecular underpinnings of the specialization of *M. oryzae* to japonica and indica varieties in Yuanyang. Our work will form the basis of testable hypotheses to determine whether the molecular mechanisms described by our model represent fundamental features of the specialization of *M. oryzae* to indica and japonica rice subspecies and in other pathosystems.

## Conclusions

This work suggests that the appropriate deployment of contrasting immune systems in the field can dramatically impact pathogen populations. In their pioneer work showing that mixtures can produce resistance, Zhu et al. (*Zhu et al., 2000*) reported a field situation where inter-cropping rice varieties dramatically reduced blast severity levels. Quite interestingly, this work involved a japonica variety (including Huang Pi Nuo, a.k.a Huang Ke Nuo) and two modern hybrid indica varieties (*Zhu et al., 2004*). We propose that part of the observed reduction of disease in this seminal work could be due to mechanisms similar to the ones uncovered in the Yuanyang terraces. The different types of resistance factors deployed may have been exposed to specialized pathogen populations, whose reduced virulence on their non-native alternate hosts would have reduced the global disease burden.

## Materials and methods

### *Magnaporthe oryzae* sampling and genotyping

Rice blast lesions identified in the field were put under 100% humidity. The resulting fungal colonies were transferred to sterile medium and single spores were isolated, DNA was extracted and analyzed using microsatellite markers according to Saleh et al (*Saleh et al., 2014*).

### Plant growth, fungal inoculations and chitin treatments

Plants and *Magnaporthe oryzae* were grown as described in Berruyer et al (*Berruyer et al., 2003*). Fungal spores (50,000 spores/mL) were inoculated by spray after three weeks (fourth leaf stage) and symptoms measured seven days after inoculation. Resistance (R) or susceptibility (S) scores were established as in Gallet et al (*Gallet et al., 2016*). The *cebip* mutant used is in the Nipponbare background (*Delteil et al., 2012*). For Avr-Effector diagnostics (*Figure 2B*), we used the rice lines described in Berruyer et al (*Berruyer et al., 2003*) that allow the identification of virulence and to some extent AVR functions. For chitin treatment, three week-old plants were sprayed with 0.02%

tween 20 (mock) or 100 µg/mL of chitin solubilized in 0.02% tween 20. The experiment was repeated four times. This chitin contains 2 to 8-mers of oligosaccharide (YSK, Yaizu Suisankagaku Industry, Japan). The third last fully expanded leaves were harvested 24 hr after treatment for gene expression analysis.

## Population genetic analyses

We used DAPC and neighbor-joining analysis of genetic distances to infer population subdivision. For both DAPC, we retained the first 20 principal components, and the first six discriminant functions. Genetic distances were computed as the proportion of shared alleles using a custom-made script, and the neighbor-joining was computed using the neighbor program from the Phylip package (http://evolution.genetics.washington.edu/phylip/progs.data.dist.html). All analyses were carried out on clone corrected datasets (i.e. on datasets for which a single representative was kept for multilocus microsatellite genotypes represented multiple times). The index of association rd is a measure of multilocus linkage disequilibrium ranges from 0 (complete panmixia) to 1 (strict clonality) (*Agapow and Burt, 2001*). The rd statistic and other summary statistics of genetic variability were computed for all clusters using a custom-made script. Significance of rd values was established by comparing the observed values with the distributions obtained by 1000 randomizations.

## Transgenic isolates

The multi-virulent isolate Guy11 was transformed with plasmids carrying *AVR-Pia* or *Avr-Pii* (*Yoshida et al., 2009*). For each *Avr*, the corresponding empty vector was used to build control strains. Single spores from transgenic events selected on the adapted antibiotic were isolated and amplified. The functionality of each *Avr* gene was tested with an inoculation of the rice differential line carrying the corresponding *Pi* gene.

## Gene expression analysis

RNA was extracted from either healthy or inoculated leaves and cDNA produced as in Delteil et al (*Delteil et al., 2012*). The primers for the rice marker genes used were previously shown to work in indica and japonica background (*Vergne et al., 2010*). Each sample consisted of at least eight plants randomly chosen and for each condition, three to four independent samples were analyzed to build the mean expression. All expression data were normalized using the expression of the constitutive Actin gene.

## Acknowledgements

We are thankful to Romiti-Michel Corinne for laboratory management. This work was supported by the INRA project 'Riz Eternel' (SPE and BAP divisions and SMaCH metaprogram). We thank S Ravel for retrieving *Pi* gene sequences and S. Duan for mapping the genomic reads. The authors thank BGI-Shenzhen, for generously offering free HiSeq X Ten sequencing platform data, with special thanks to M Wang in the bioinformatics analysis. We also thank H Finney for proofreading this manuscript.

## Additional information

### Funding

| Funder | Grant reference number | Author |
| --- | --- | --- |
| Institut National de la Recherche Agronomique | SMAcH project (Riz Eternel) | Isabelle Meusnier<br>Henri Adreit<br>Aurélie Ducasse<br>François Bonnot<br>Thomas Kroj<br>Elisabeth Fournier<br>Didier Tharreau<br>Pierre Gladieux<br>Jean-Benoit Morel |

The funders had no role in study design, data collection and interpretation, or the decision to submit the work for publication.

## Author contributions

JL, HH, Acquisition of data, Analysis and interpretation of data, Drafting or revising the article; IM, HA, AD, Acquisition of data, Analysis and interpretation of data; FB, Analysis and interpretation of data; LP, XH, Acquisition of data; TK, EF, DT, PG, Conception and design, Analysis and interpretation of data, Drafting or revising the article; J-BM, Conception and design, Acquisition of data, Analysis and interpretation of data, Drafting or revising the article

## Author ORCIDs

Jean-Benoit Morel, http://orcid.org/0000-0003-1988-956X

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
