## [Decision Letter]

Thank you for submitting your article "Pathogen effectors and plant immunity play a central role in specialization of the rice blast fungus to rice subspecies" for consideration by *eLife*. Your article has been favorably evaluated by Detlef Weigel as the Senior Editor and two reviewers: Jian-Min Zhou (Reviewer #1), who is a member of our Board of Reviewing Editors, and Sophien Kamoun (Reviewer #2).

The reviewers have discussed the reviews with one another and the Reviewing Editor has drafted this decision to help you prepare a revised submission.

This is an interesting study that focuses on the structure and virulence of rice blast population in a traditional agroecosystem (Yuanyang). The originality of this study is that they studied paired fungus/cultivars. They surprisingly discovered a highly structured pathogen population and went to investigate why IB (Indica-borne) isolates do not infect japonica cultivars. They propose that japonica cultivars have fewer *Pi* resistance genes and more elevated levels of basal resistance. Pathogen strains that carry fewer effectors are less competitive on japonica cultivars since they cannot overcome the higher levels of basal resistance. They ascribed the effect to primarily one effector *Avr-Pia*, which can enhance virulence levels. The work is original and important, but some of the key findings are not fully supported by data.

Major comments:

1) The suggestion that japonica cultivars have stronger "basal resistance" is too preliminary. This is essentially based on study of the expression of 5 defense-related genes in two indica cultivars and three japonica cultivars following *M. oryzae* inoculation. It is unacceptable averaging out expression for multiple genes as in Figure 3. The authors should process every single gene/data separately. Do the authors mean immunity triggered by pattern-recognition receptors when they say "basal resistance"? Simple inoculations with the "virulent isolates" do not necessarily mean one is looking at PRR-mediated defenses. In Figure 3—figure supplement 3 they showed responsiveness to chitin, which measures PRR-mediated defenses. Unfortunately, the authors did not include indica cultivars.

2) The claim that *Avr-Pia* contributes to virulence in this agri-system is not well supported. The authors introduced *Avr-Pia* into the laboratory isolate GUY11 and inoculated onto several local cultivars. It enhanced disease lesions only on a single cultivar: Huang Pi Nuo but not other cultivars. While this experiment shows that *Avr-Pia* can enhance virulence, GUY11 is not a local strain. A better experiment would be to knock-out this gene from local isolates to demonstrate that this gene indeed contributes to virulence.

3) The authors conclude that japonica cultivars have fewer *Pi* genes than indica cultivars. Figure 2 showed a single japonica cultivar and two indica cultivar. The data cannot be generalized. Also, why were two methods used to estimate the number of *Pi* genes (each for parts of the cultivars and then combine)? A unified method (sequence analysis) should be used for all the estimation.

4) Throughout the study there is too much reliance on averages and representation of data as histograms. The distribution of the data and variance is important here (example, number of effectors per isolate).

5) Japonica cultivars only form 2% of the plants grown in this agrosystem. How does this affect the population analyses and conclusions? Would they expect the same population structure if it was the other way around with Japonica at 98%?

6) How a Japonica cultivar with functional RGA4/RGA5 gene would perform?

---

## [Author Response]

*Major comments:*

*1) The suggestion that japonica cultivars have stronger "basal resistance" is too preliminary. This is essentially based on study of the expression of 5 defense-related genes in two indica cultivars and three japonica cultivars following M. oryzae inoculation.*

We have extended our analysis to 27 defense-related genes known to be regulated during infection. We believe that the conclusions are now much more solid. As for the limited number of cultivar tested, we modified the text to avoid over-statement. However, the four varieties studied represent the vast majority of the grown surface in Yuanyang and our conclusions on the local adaption holds true for this area of co-cultivation of indica and japonica.

*It is unacceptable averaging out expression for multiple genes as in Figure 3. The authors should process every single gene/data separately.*

We have changed the figures to fulfill this comment. There are now several figures (Figure 3, Figure 3—figure supplement 5, Figure 3—figure supplement 6) that make the point that induction of defense is higher in Yuanyang varieties inoculated by Guy11. We also have extended this observation to two other isolates (Figure 3—figure supplement 7).

*Do the authors mean immunity triggered by pattern-recognition receptors when they say "basal resistance"? Simple inoculations with the "virulent isolates" do not necessarily mean one is looking at PRR-mediated defenses.*

We agree that inoculation with “virulent isolates” is not exactly measuring PRR-mediated response; this is why we have done experiments with chitin. We have tried to make this point clearer in the text (subsection “Elevated basal immunity in Yuanyang japonica varieties”, first and last paragraphs).

*In Figure 3—figure supplement 3 they showed responsiveness to chitin, which measures PRR-mediated defenses. Unfortunately, the authors did not include indica cultivars.*

We made no hypothesis on the responsiveness of indica varieties to chitin as this is not a pre-requisite to our model. Indeed, the fact that indica varieties could have high or low levels of responsiveness to chitin would not impact the anyway high level of effector-triggered immunity in these varieties. This is why we did not include indica varieties in this analysis.

*2) The claim that Avr-Pia contributes to virulence in this agri-system is not well supported. The authors introduced Avr-Pia into the laboratory isolate GUY11 and inoculated onto several local cultivars. It enhanced disease lesions only on a single cultivar: Huang Pi Nuo but not other cultivars.*

We have now included data from the only other japonica variety Nuo Gu which shows the same trend than the Huang Pi Nuo one. We have modified the text at the end of the Introduction, the end of the Results section, and in the subsection “Effectors are required for host specialization in the rice/rice blast system” to be more specific; the Abstract has also been modified accordingly.

*While this experiment shows that Avr-Pia can enhance virulence, GUY11 is not a local strain. A better experiment would be to knock-out this gene from local isolates to demonstrate that this gene indeed contributes to virulence.*

Since numerous effectors in JB isolates could have redundant functions, we reasoned that deleting *AVR-Pia* or *AVR-Pii* from these isolates may not strongly affect virulence. We therefore decided to test the role of these effectors by transferring them into the *AVR* depleted strain Guy11. This comment was included in the Result section (subsection “AVR-Pia contributes to virulence toward some japonica rice varieties”).

*3) The authors conclude that japonica cultivars have fewer Pi genes than indica cultivars. Figure 2 showed a single japonica cultivar and two indica cultivar. The data cannot be generalized.*

We fully agree with this comment and we have stated whenever required that our data only apply to locally grown indica and japonica varieties. For instance see subsection “Differentiation of *Magnaporthe oryzae* populations infecting indica or japonica rice in Yuanyang”, last paragraph and subsection “Effectors are required for host specialization in the rice/rice blast system”.

*Also, why were two methods used to estimate the number of Pi genes (each for parts of the cultivars and then combine)? A unified method (sequence analysis) should be used for all the estimation.*

Since the initial submission, we have generated whole-genome sequence of the four indica and japonica varieties studied in this work. We used this data to fully picture the situation about 19 cloned Pi resistance genes. The picture is now even stronger than before as we discovered many stop codons in Pi genes in japonica, further reducing the number of functional genes in these varieties.

*4) Throughout the study there is too much reliance on averages and representation of data as histograms. The distribution of the data and variance is important here (example, number of effectors per isolate).*

Whenever possible, we have adopted a presentation where all data are visible.

*5) Japonica cultivars only form 2% of the plants grown in this agrosystem. How does this affect the population analyses and conclusions? Would they expect the same population structure if it was the other way around with Japonica at 98%?*

We can only speculate an answer to this question. However, the reviewer stresses a point that is very relevant: despite the large size of the indica-borne isolates, we almost never found these isolates on japonica rice, which is the starting point. This point is raised in the manuscript (subsection “Plant immunity shapes effector complements and specialization to hosts of the rice blast pathogen”).

*6) How a Japonica cultivar with functional RGA4/RGA5 gene would perform?*

Such a variety has been tested in this study (Figure 2—figure supplement 4; Aichi Asahi) and was resistant to almost JB isolates (as they contain *AVR-Pia*) but susceptible to all Indica-born isolates. In the field, such a variety is expected to be highly resistant to JB isolate since effector-triggered immunity is likely epistatic to effector-triggered susceptibility.